# A New Insight into MYC Action: Control of RNA Polymerase II Methylation and Transcription Termination

**DOI:** 10.3390/biomedicines11020412

**Published:** 2023-01-30

**Authors:** Fiorella Scagnoli, Alessandro Palma, Annarita Favia, Claudio Scuoppo, Barbara Illi, Sergio Nasi

**Affiliations:** 1IBPM—CNR, Biology and Biotechnology Department, Sapienza University, 00185 Rome, Italy; 2Translational Cytogenomics Research Unit, Bambino Gesù Children’s Hospital, IRCCS, 00146 Rome, Italy; 3Institute for Cancer Genetics, Columbia University, New York, NY 10032, USA

**Keywords:** Myc, RNA Polymerase II, transcription termination, cancer cell transcriptome

## Abstract

MYC oncoprotein deregulation is a common catastrophic event in human cancer and limiting its activity restrains tumor development and maintenance, as clearly shown via Omomyc, an MYC-interfering 90 amino acid mini-protein. MYC is a multifunctional transcription factor that regulates many aspects of transcription by RNA polymerase II (RNAPII), such as transcription activation, pause release, and elongation. MYC directly associates with Protein Arginine Methyltransferase 5 (PRMT5), a protein that methylates a variety of targets, including RNAPII at the arginine residue R1810 (R1810me2s), crucial for proper transcription termination and splicing of transcripts. Therefore, we asked whether MYC controls termination as well, by affecting R1810me2S. We show that MYC overexpression strongly increases R1810me2s, while Omomyc, an MYC shRNA, or a PRMT5 inhibitor and siRNA counteract this phenomenon. Omomyc also impairs Serine 2 phosphorylation in the RNAPII carboxyterminal domain, a modification that sustains transcription elongation. ChIP-seq experiments show that Omomyc replaces MYC and reshapes RNAPII distribution, increasing occupancy at promoter and termination sites. It is unclear how this may affect gene expression. Transcriptomic analysis shows that transcripts pivotal to key signaling pathways are both up- or down-regulated by Omomyc, whereas genes directly controlled by MYC and belonging to a specific signature are strongly down-regulated. Overall, our data point to an MYC/PRMT5/RNAPII axis that controls termination via RNAPII symmetrical dimethylation and contributes to rewiring the expression of genes altered by MYC overexpression in cancer cells. It remains to be clarified which role this may have in tumor development.

## 1. Introduction

In this article, we present data on physical and functional interactions among MYC, RNAPII, and PRMT5.

### 1.1. MYC

Deregulated expression of the transcription factor MYC is a crucial event in a wide variety of cancer cells. Deregulation—usually due to overexpression—affects transcriptional networks controlling key cell functions such as proliferation, stemness maintenance, metabolism, and pre-mRNA splicing. Therefore, it is not surprising that MYC inhibition represents an attractive strategy against many cancer types MYC dimerizes with MAX through the bHLHZip domain. MYC-MAX dimers bind to promoter proximal regions of genes and interact with the transcription machinery. It is unclear whether MYC mainly acts as a universal amplifier of the transcriptional program running within a given cell, a specific activator of a distinct set of target genes, or both [1,2,3,4,5,6,7].

### 1.2. MYC and RNAPII

Besides MAX, MYC can associate with a wide range of proteins, including CDK9—the kinase subunit of p-TEFb—and SPT5, which bind to the RNAPII carboxyl terminal domain (CTD). The RNAPII CTD is crucial for the control of transcription, mRNA processing, and nucleosome modifications. It contains 52 tandem repeats of the consensus sequence N-Tyr1-Ser2-Pro3-Thr4-Ser5-Pro6-Ser7-C as well as several Arg (R) and Lys (K) residues that are targets of post-translational modifications—phosphorylation, ubiquitination, methylation—which regulate distinct steps of transcription [8,9,10,11]. The TFIIH complex and p-TEFb mediate, respectively, phosphorylation of Serine 5 and 2 (Ser5P, Ser2P) in the RNAPII CTD, two modifications that control early transcription events. During transcript elongation, Ser5P levels decrease and Ser2P levels increase [8,12,13,14,15]. Via the interaction with SPT5 and p-TEFb, MYC enhances transcriptional pause release and elongation [13,16,17,18,19,20].

### 1.3. Omemee

One of the best tools to interfere with Myc function, avoiding the use of gene KO techniques, is Omomyc, an MYC-derived 90 amino acid mini-protein. Omomyc constrains MYC activity in cancer cells and in vivo models by directly affecting MYC binding to DNA and MYC protein–protein interactions, as shown by our previous works [21,22,23,24]. Omomyc has anticancer properties in a wide variety of tumors. Indeed, in *KRAS*-mutated Myc overexpressing lung cancer cells, Omomyc induces cell death [25], whereas in gliomas it slows tumor growth, inducing apoptosis and the occurrence of multinucleated cells that undergo either growth arrest or death by mitotic catastrophe [26]. 

Therefore, to gain insights into deregulated pathways in Myc overexpressing cancer cells, Omomyc represents an extremely useful opportunity. 

### 1.4. MYC, RNAPII, and PRMT5

We reported that MYC associates with PRMT5, an arginine methyl-transferase that monomethylates and symmetrically dimethylates histone and non-histone proteins [27,28]. 

PRMT5 acts on transcription through several histone methylations, which can either activate or repress transcription. In particular, repressor activity implicates symmetrical dimethylation of R8 and R3 on histones H3 and H4, respectively, which lead to enhanced DNA methylation and chromatin compaction. The functional interaction between MYC and PRMT5 occurs at multiple levels. Indeed, MYC upregulates PRMT5—which methylates Sm proteins—and the core small nuclear ribonucleoprotein particle assembly genes in lymphoma cells, maintaining splicing fidelity during lymphomagenesis [29]. Moreover, MYC enhances PRMT5-dependent symmetrical dimethylation of H4 on R3 (H4R3me2s); PRMT5, in turn, regulates MYC activity [27,28]. Moreover, PRMT5 participates in transcriptional repressor complexes that include Histone Deacetylases (HDACs) and/or DNA methyl transferases (DMNTs) [30,31,32,33,34,35,36]. By dimethylating R2 on histone H3, instead, PRMT5 functions as a transcriptional activator [37,38]. Thus, it is not unexpected that PRMT5 has multiple roles in cell biology—such as the regulation of neural differentiation, Golgi trafficking, and stemness maintenance—and is deregulated in a variety of tumors [39,40,41,42,43]. Notably, PRMT5 inhibitors have gained a strong interest for developing new treatments for cancer [43,44,45,46]. Interestingly, RNAPII as well is a direct PRMT5 target [11]. In particular, PRMT5 was shown to symmetrically dimethylate RNAPII CTD arginine residue R1810 (R1810me2s). This modification was shown to recruit the survival motoneuron protein (SMN) to RNAPII elongation complexes. SMN, in turn, binds to a DNA-RNA helicase that resolves R-loops in TTS (transcription termination sites) and is critical for proper termination and splicing of RNAPII transcripts [11].

The findings that PRMT5 has a key role in transcription termination and interacts with MYC strongly suggested that MYC might be involved in transcription termination as well, through PRMT5. In the present work, we employ MYC inhibition by shRNA and by Omomyc to investigate MYC’s role in regulating RNAPII post-translational modifications, RNAPII distribution, and gene expression in cancer cells. Our study is suggestive of another control level exerted by MYC, which makes increasingly evident its role as master controller of gene expression, involved in RNAPII activation, elongation, termination, and pre-mRNA splicing.

## 2. Materials and Methods

### 2.1. Cell Lines, Culture, and Treatments

Brain Tumor 168 (BT168) glioblastoma stem cells (GSC) have been previously described by De Bacco et al. 2012 [47]. Cells were grown as neurospheres in serum-free medium, Dulbecco’s Modified Eagle Medium/F-12 (DMEM/F12; SIGMA; St. Louis, Mo, USA) supplemented with B-27™ Supplement, 1% penicillin/streptomycin, 2 mM Glutamine, 10 ng/mL EGF and bFGF (Thermo Fisher Scientific; Waltham, MA, USA). Burkitt’s lymphoma Ramos cells have been previously described [48]. Cells were cultured in RPMI-1640 medium supplemented with 10% Fetal Bovine Serum (FBS; Thermo Fisher Scientific), 1% penicillin/streptomycin, 2 mM Glutamine. HEK293T cells were cultured in DMEM (SIGMA, St. Louis, Mo, USA), supplemented with 10% FBS, 2 mM Glutamine and penicillin/streptomycin. Cells harboring a doxycycline-inducible FlagOmomyc were obtained by lentiviral infection. BT168FO and RamosFO cells were treated, respectively, with 0.25 μg/mL and 0.1 μg/mL doxycycline (SIGMA, St. Louis, Mo, USA) to induce Omomyc. BT168shMYC1# cells were obtained by transduction with an inducible lentivirus expressing a short hairpin RNA for MYC and treated with 0.25 μg/mL doxycycline. HEK293T cells were treated with the 5 μM of the PRMT5 inhibitor EPZ01566 (1:1000, SIGMA, St. Louis, Mo, USA). Cells were harvested 48 h after treatment and the inhibition of PRMT5 activity was tested with immunoblots for H4R3me2s.

### 2.2. Lentiviral Infection

The lentiviral plasmid pSLIK-FO has already been described [24]. The lentiviral plasmid pSLIK-shMYC1# (sh sequence TGCTGTTGACAGTGAGCGAAAGATGAGGAAGAAATCGATGTAGTGAAGCCACAGATGTACATCGATTTCTCCTCATCTTCTGCCTACTGCCTCGGA) was engineered by cutting pSLIK-FO using PacI and SnaBI to cut away the Gateway platform. The fragment PacI-SnaBI was purified. PCR from GEPIR (all-in-one shRNA-vector; [49]) for the TRE3G-EGFP-mir30E band inserted the SnaBI and PacI sites. The fragment TRE3G-EGFP-mir30E was purified and cloned in the pSLIK-PacI-SnaBI vector. pSLIK-SnaBI-mir30E-PacI was cut with SnaBI for re-inserting RRE and the Flag sequence. The final vector pSLIK-shMYC co-express the hygromycin resistance gene and Tet-transactivator rtTA3. Lentiviruses were prepared by co-transfecting HEK293T cells with pSLIK-Flag-Omomyc and packaging plasmids PLP1, PLP2, and pMD VSV-G diluted in Opti-MEM (Thermo Fisher Scientific, Waltham, MA, USA). The medium was removed after 12–24 h and replaced with 4 mL of fresh growth media. Supernatants were collected every 24 h between 48 and 72 h after transfection, pulled together, and concentrated by ultracentrifugation in a Beckman SW-28 rotor for 2 h at 25,000 rpm, 4 °C. For infection, 2–5 × 10^5^ cells were seeded in 35 mm dishes and infected the following day in the presence of 4 µg/mL polybrene. BT168FO cells were selected with 50–200 μg/mL hygromycin B (SIGMA, St. Louis, Mo, USA), Ramos FO cells were selected with 400–800 µg/mL higromycin B. After selection, Flag-Omomyc and shMYC expression were assessed by western blots.

### 2.3. Transfection

FlagOmomyc (pCbsFlagOmomyc), FlagMYC (pCbsFlagMYC), and pSLIKshMYC plasmids were transfected using Lipofectamine 2000 (Thermo Fisher Scientific; Waltham, MA, USA) according to the manufacturer’s instructions. A total of 50 nM PRMT5-siRNA or control siRNA (Dharmacon, Lafayette, CO, USA, SiRNA-SMART pool) were transfected with DHARMAFect transfection reagent (Dharmacon, Lafayette, CO, USA) according to the manufacturer’s instructions. Cells were harvested 48 h after transfection.

### 2.4. Immunoprecipitation

A total of 10–20 × 10^6^ cells were lysed on ice in 140 mM NaCl, 10 mM Tris pH 7.6–8.0, 1% Triton, 0.1% sodium deoxycoholate, 1 mM EDTA, containing protease inhibitors (Roche) and benzonase (SIGMA, St. Louis, Mo, USA) for 25′ by vortexing and forcing them through a 27-gauge needle, at least 10 times [16]. After centrifuging at 13,000 rpm for 15 min at 4 °C, the supernatant was incubated with 25–30 µL of protein G dynabeads (Thermo Fisher Scientific, Waltham, MA, USA) conjugated with 4 μg of antibodies for 4 h overnight (O/N). The samples were washed 3 times with lysis buffer and boiled in Laemmli buffer. To detect R1810me2s modification on RNA polymerase II (RNAPII), RNAPII immunoprecipitated samples were treated with alkaline phosphatase (Roche) (5 μL) at 37 °C for 30′ before boiling [11].

### 2.5. Immunoblotting

Proteins were resolved in 6-8-10 or 12% polyacrilammide gels and transferred to PVDF (Bio-Rad, Hercules, CA, USA) or nitrocellulose membranes (GE Heathcare, Chicago, IL, USA) for 2 h at 250 mA on ice or overnight at 30 V. Filters were blocked in PBS/0.1% Tween-20 (SIGMA, St. Louis, Mo, USA) added with 10% non-fat dry milk, for 1 h30′ at room temperature (RT). Primary antibodies were incubated O/N at 4 °C, according to the concentration recommended by the manufacturer, in PBS/0.1% Tween-20 plus 2.5–5% non-fat dry milk. After three 10′ washes, filters were incubated for 1 h at RT with either goat-anti rabbit (1:5000) or goat-anti mouse (1:2000) horseradish peroxidase (HRP)-conjugated secondary antibodies (Merck). Blots were developed using SuperSignal West Pico or Femto Maximum Sensitivity Chemiluminescent Substrate (Thermo Fisher Scientific, Waltham, MA, USA). Images were captured with a Chemidoc XRS+ (Bio-Rad, Hercules, CA, USA) and quantified using ImageJ software. To avoid possible errors in western blots band quantification, we used a background subtraction method described in Gallo-Oller et al., J Immunol Methods, 2018 [50]. Anti-MYC (9E10, cat. sc-40; N-262, cat. sc-764), anti-CDK9 (cat. sc-376646), anti-RNAPII (8WG16, cat. sc-56767) antibodies were from Santa Cruz Biotechnologies, (Dallas, TX. USA), anti-H4R3me2s (cat. ab5823), anti-PRMT5 (cat. ab109451), and anti-RNA polymerase II CTD repeat YSPTSPS (phospho S2; cat. ab24758) antibodies were from Abcam (Cambridge, UK); anti-Flag antibody (cat. F1804) and anti-β-Actin-peroxidase (cat. A3854) were from SIGMA (St. Louis, Mo, USA). Anti-R1810me2s was courtesy of J.F. Greenblatt’s lab—University of Toronto [11]. 

### 2.6. Chromatin Immunoprecipitation (ChIP), ChIP-seq and RNA-seq

Samples for ChIP and ChIP-seq assays were prepared and analyzed according to Myers Lab ChIP-seq Protocol v041610 (http://myers.hudsonalpha.org/documents/, first accessed on 1 November 2011) by using MAGnify Chromatin Immunoprecipitation System protocol (Thermo Fisher Scientific, Waltham, MA, USA). Antibodies used: MYC (sc-764Z, Santa Cruz Biotechnologies, Dallas, TX. USA), MAX (sc-197X, Santa Cruz, Dallas, TX. USA), RNAPII (sc-899X, Santa Cruz, Dallas, TX. USA), RNAPII phospho Ser5 (ab5131, Abcam, Cambridge, UK), RNA Pol II phospho Ser2 (ab24758, Abcam, Cambridge, UK and 3E19, Active Motif, Vinci-Biochem, FI, Italy), Flag (F1804, Sigma, St. Louis, Mo, USA). For RNA-seq, 2µg total RNA purified by PureLinkRNA Mini Kit (Thermo Fisher Scientific, Waltham, MA, USA) was used. ChIP-seq and RNA-seq libraries were prepared at Istituto di Genomica Applicata (IGA; www.appliedgenomics.org/ accessed on 13 December 2022) according to Illumina TruSeq DNA and TruSeq RNA Sample Preparation Guides. Samples were sequenced through Illumina HiSeq 2000 e 2500.

### 2.7. Data Processing and Bioinformatics Analysis

Data were processed as described by Galardi et al., 2016 [24]. For ChIP-seq analysis, 50-bp reads were mapped to hg19 human reference genome (UCSC Genome Browser) using Bowtie [50] version 0.12.7 allowing three mismatches; reads with multiple best matches were discarded. Peak calling was through MACS [51] 1.4.2 with a 10-4 *p*-value cut-off. The RefSeq transcript annotation of hg19 was used for computing intersections between peaks and promoters. Binding enrichment to promoters was calculated by the normalized number of ChIP-seq reads as Reads Per Million (RPM). In the case of multiple TSSs, those with the highest enrichment were chosen. Motif enrichment analysis was performed by Pscan-ChIP [52]. Seqminer v.1.3.3 was used to calculate distribution around TSSs. The RAP RNA-Seq pipeline (https://bioinformatics.cineca.it/rap/, first accessed on 1 November 2011)—including quality controls, adaptor trimming and masking of low-quality sequences, tophat2, bowtie, and CuffLinks 2.2—was used to reconstruct the transcriptome (hg19 reference) and calculate expression values as FPKM (Fragment per Kilobase Million per genes). Data have been analyzed using the DESeq2 R package [53], considering genes with a FPKM > 0. Differentially expressed genes between treated (24 h and 48 h) and untreated samples with an adjusted *p*-value < 0.05 were taken as up-regulated (log2 fold change > 0) or down-regulated (log2 fold change < 0). Up- and down-regulated genes were separately used for Gene Ontology Enrichment Analysis using EnrichR [54] and Gene Set Enrichment Analysis using GSEA [55]. Data and figures were further analyzed using in-house R scripts and the Perseus tool [56]. Comparisons between MYC and Omomyc occupancy and gene expression (FPKM) were performed by calculating the average values for groups of 100 genes (bins) and correlated by a scatter diagram. The linear regression model was used to assess the correlation between transcript levels in NODOX versus DOX cells. RNAPII distribution, at TTS versus TSS regions, was evaluated using ChIP-seq data. Density reads, counted as RPKM, for each gene, at promoter (1500 nt) and termination (4200 nt) regions were calculated by dividing the number of reads by the total number of reads obtained from each sequencing per condition (−DOX and +DOX), and by the length of the features. Data were normalized by their input. Gene Set Enrichment Analysis (GSEA, http://www.broad.mit.edu/gsea/index.html, first accessed on 1 November 2011) was used to determine whether an a priori defined set of genes shows statistical significance, according to the differences between −DOX and +DOX experimental conditions (phenotypes). In detail, the RNA-Seq dataset files—consisting of experiments in triplicate for each time point of DOX treatment—containing two labeled phenotypes (−DOX and +DOX) were prepared in TXT format: −DOX included all 0 h time points (1° phenotype), while +DOX included from 4 h to 48 h of DOX treatment (2° phenotype). The expression dataset was compared with several gene sets either exported from the GSEA-MsigDB database or homemade. The gene sets contained the gene set name and the list of included genes. A gene set file was in GMX or GMT format. GSEA software calculated an enrichment score (ES) describing the degree to which a gene set was overrepresented at the extremes (top or bottom) of the entire ranked list of the data set—where genes are ranked according to the expression difference between −DOX and +DOX conditions. The Enrichment Score (ES) was calculated by walking down the list. The value statistically increased when it found genes present in the gene set and decreased when genes were not present. The magnitude of the ES was dependent on the correlation of each gene with the phenotype. The proportion of false positives was evaluated by calculating the False Discovery Rate FDR-q value. Refseq IDs were mapped onto gene symbols using the biormaRt R tool [57]. Analyses were performed on the mean value of promoter and termination sites for each condition, calculated as the mean value of two independent replicates normalized per million of mapped reads. Correlation analysis between RNA sequencing and Chip sequencing was performed using the log2 fold change value for RNA-seq and the mean promoter/termination site for Chip-seq. RNAPII density was calculated as the ratio between the mean termination site and the mean promoter site values.

### 2.8. Statistical Analysis

Statistical analyses were performed by using GraphPad Prism version 5.0d and Excel (Microsoft Excel, version 2018). All histograms represent the mean ± SEM of data obtained in 3 or more independent experiments. Statistical significance was determined by one-way repeated-measures ANOVA or a paired t-test. The box plot *p*-values were calculated by paired Wilcoxon signed-rank tests. Regression lines were estimated using linear regression models. For genomic data, differential expression was assessed by CuffDiff2, as well as by Fold-Change thresholds, and Gene Set Enrichment Analysis (GSEA: www.broadinstitute.org/gsea/, first accessed on 1 November 2011) subdividing MYC targets and non-MYC targets into groups of 500 genes.

## 3. Results

### 3.1. MYC Induces RNAP II Symmetrical Dimethylation of R1810

To gain insights into a potential MYC role in transcription termination, we investigated its ability to influence symmetrical dimethylation of RNAPII R1810, a key modification that is catalyzed by PRMT5 and regulates termination, by gain and loss of function experiments. To this end, we employed Omomyc, an MYC-specific shRNA, the PRMT5 activity inhibitor EPZ015666 and a smart pool of siRNAs for PRMT5 (siP5). We transfected HEK293T recipient cells with a FlagMYC expression construct, in the presence or absence of an MYC shRNA and FlagOmomyc expression plasmids. Following or not immunoprecipitation with an RNAPII antibody, the protein extracts were analyzed by western blotting with antibodies specific for PRMT5, SMN, R1810me2s, and RNAPII (Figure 1a,b). We found that ectopic MYC expression strongly enhanced RNAPII R1810me2s, which was almost totally abolished by co-transfection with MYC-specific shRNA, as shown by the immunoprecipitations and the related densitometry histogram in Figure 1a,b. Co-transfection of the Omomyc expression plasmid strongly blunted the R1810 symmetrical dimethylation increase caused by MYC ectopic expression. To verify that PRMT5 was required for the R1810me2s increase caused by overexpressed MYC, HEK293T cells were transfected with the FlagMYC vector and treated or not with the PRMT5 inhibitor EPZ015666 24 h after transfection. The PRMT5 inhibitor prevented the MYC-dependent increase of R1810 dimethylation (Figure 1c, middle). The same result was obtained when FlagMYC overexpressing HEK293T cells were first transfected with a siP5 smart pool siRNA, blunting PRMT5 expression (Figure 1d).

### 3.2. MYC Inhibition Decreases RNAPII Symmetrical Dimethylation in Cancer Cells

To evaluate MYC action on RNAPII symmetrical dimethylation, we took advantage of two cancer cell types with high MYC basal levels: the glioblastoma stem cell (GSC) line named BT168 and the Burkitt’s lymphoma cell line Ramos, in which the *myc* gene is under control of the immunoglobulin heavy chain promoter [47,48]. These cancer cell lines are an extremely useful model to study the effect of MYC at the molecular layer which, through the execution of deregulated gene expression programs, consequently impacts cell behavior. In order to investigate whether MYC may play a role in modulating RNAPII transcription termination activity in MYC overexpressing cancer cells, the two cell lines were stably transduced with a doxycycline-inducible, FlagOmomyc (FO) expressing lentivirus [24]. No remarkable changes were observed in cell morphology upon Omomyc induction (not shown). As shown in Figure 2a,c, doxycycline treatment of BT168FO cells and RamosFO cells led to a strong reduction of RNAPII R1810me2s, which in these cell systems was basally detected with respect to HEK293T cells (Figure 1) where MYC protein level is low. The same result was obtained upon doxycycline treatment of BT168 cells stably transduced with a lentivirus expressing a doxycycline-inducible shRNA against MYC (Figure 2b). In both cell types, the decrease of RNAPII symmetrical dimethylation was paralleled by a decrease of SMN binding to RNAPII, as expected.

### 3.3. MYC Inhibition Decreases RNAPII Serine 2 Phosphorylation and Modulates MYC and MAX Expression in Cells Expressing High MYC Levels

The transition between initiation and productive elongation is elicited by Ser5 phosphorylation in RNAPII CTD, followed by the elongation-specific Ser2 phosphorylation. MYC binds to p-TEFB and regulates transcriptional pause release. Therefore, we asked whether Omomyc might also affect Ser2 phosphorylation, thus influencing the transcription elongation rate and mRNA expression. 

First, HEK293T cells were transfected with FlagMYC, FlagOmomyc, and pSLIKshMYC plasmids, either alone or in combination. Immunoblotting analyses with RNAPII Ser2P-specific antibodies showed that MYC ectopic expression strongly enhanced Ser2 phosphorylation (Figure 3a, left), as expected. In parallel, MYC overexpression caused an increase in the p-TEFB component CDK9 (Figure 3a, right). Omomyc expression as well enhanced Ser2 phosphorylation. In co-transfection experiments, instead, Omomyc caused a significant reduction of the levels of RNAPII Ser2P phosphorylation observed in the presence of over-expressed MYC (Figure 3a, left). Ser2P reduction was also obtained by cotransfection with MYC shRNA (Figure 3a), as expected. In BT168FO cells, RNAPII Ser2P decreased upon Omomyc induction (Figure 3b). MYC overexpression in cancer cells contributes to the formation of a high number of MYC/MAX dimers that invade transcriptionally active chromatin sites [2,3,5,7,58]. Given the ability of Omomyc to interfere with the formation of MYC protein complexes [23], we asked whether Omomyc might affect the expression of MYC and MAX. We found that Omomyc induction in BT168 and Ramos cells decreased the expression of MYC protein, in parallel with an increase in MAX (Figure 3c). In HEK293T cells—which have low MYC levels compared to cancer cells such as Ramos and BT168—Omomyc did not significantly influence MYC expression (Figure 1A, middle).

### 3.4. Omomyc Specifically Represses Direct MYC Target Genes

To further examine how Omomyc may influence the glioblastoma stem cell transcriptome, we measured by RNA-seq the mRNA output changes consequent to 24 and 48 h Omomyc induction in BT168FO cells. Strongly expressed MYC target genes (FPKM ≤ 10 in at least one condition) are shown in Figure 4a,b. As expected, the number of differentially expressed genes was higher in cells treated longer (Appendix A). A 48 h treatment led to 2228 differentially expressed genes, 1606 of which were up-regulated and 622 were down-regulated (Figure 4a,b) and (Appendix A). 

To elucidate the meaning of differentially expressed gene sets, we performed functional enrichment analyses by means of GO, KEGG, Reactome, WikiPathways. While we found little significance at 24 h DOX treatment—probably due to the lower number of differentially expressed genes (Appendix A)—various pathways were significantly enriched after 48 h DOX treatment, coherently in the different analyses (Figure 4c). Downregulated genes were enriched for gene ontology (GO) terms related to DNA metabolic process, DNA replication, DNA repair, and cell cycle, in agreement with the view that MYC primarily acts as a transcriptional activator controlling metabolic and biosynthetic processes [18,59]. These GO terms were also correlated with MYC expression in a variety of cancer cell lines [2,19,24,59]. The pathways enriched among the upregulated genes were related to cell and amino acid metabolism, and lysosome activity. Altogether, these analyses indicate that Omomyc specifically influences the expression of genes and pathways that are regulated by MYC.

To investigate the effect of Omomyc on the expression of genes directly regulated by MYC, we resorted to the signature of 100 direct MYC targets—conserved in a variety of cancer cell lines—described by Muhar and coworkers [59]. Such *Muhar signature* genes present a high level of MYC binding to the promoter, and their expression level in a given cell line correlates well with the MYC amount in that cell line. We thus investigated whether Omomyc influenced the expression of this annotated gene set. We found that as many as 91 out of 100 *Muhar signature* genes were expressed in BT168FO cells (at FPKM > 1), almost all significantly down-regulated by Omomyc, as shown by the negative score in the Gene Set Enrichment Analysis (GSEA) and by the heatmap and profile score of the signature genes (Figure 4d,e). 

### 3.5. Omomyc Affects RNAPII Density at Promoter and Termination Sites

MYC binds to most active promoters enhancing transcription pause release and elongation [13,16]. It also increases RNAPII R1810 symmetric dimethylation (Figure 1), which regulates transcription termination [11]. Omomyc forms dimers that compete with MYC for DNA binding—causing a 50–60% reduction of MYC binding to promoters in BT168 cells [4]—and restrains RNAPII symmetrical dimethylation, involved in transcription termination. This suggested to us that Omomyc might affect RNAPII density at promoters and/or terminator regions and influence gene expression, at least partly, in this way. In order to assess the influence of Omomyc on RNAPII distribution at promoter and terminator regions, we analyzed RNAPII ChIP-seq data of BT168FO cells treated or not with Dox, focusing on the genes that displayed promoter binding by MYC and were highly expressed (FPKM ≥ 10). RNAPII density on either promoter and terminator regions of DOX treated versus control cells presented a clear linear correlation (Figure 5a,b). We found that Omomyc induction led to a strong increase (1.5–2 fold) in the amount of RNAPII bound to promoter and termination sites (Figure 5a,b). RNAPII relative density—expressed as the ratio between terminator and promoter density—was instead insensitive to Omomyc, remaining unchanged or showing only a very slight decrease in DOX treated versus untreated cells (Figure 5c).

This trend was also present in the *Muhar signature* of MYC target genes (Figure 5d–g), with higher values of RNAPII density at promoters and terminators in DOX-treated cells. As to mRNA expression levels, *Muhar signature* genes were significantly down-regulated by Omomyc induction. The correlation analysis for all MYC target genes (Figure 5, third row) did not show a strong link between mRNA expression changes and RNAPII occupancy.

## 4. Discussion

The concept of the MYC oncogene as a “traditional” transcription factor has consistently changed in the last 10 years. Indeed, MYC has been found to work during the transcription process not only as a chromatin binding factor, directly reorganizing the cancer genome, invading chromatin regulatory loci, such as promoters [3,19] and super enhancers [60] promoting—in the vast majority of cases—transcription activation, but also regulating many aspects linked to RNAPII activity such as release of transcriptional pausing, elongation of transcripts [13,16], and regulation of splicing [29]. A role of MYC in tuning transcription termination has not been unraveled so far. Termination of transcripts is a complex process that requires, in its first steps, the symmetrical dimethylation of R1810 located at the C-terminal domain of RNAPII by PRMT5 and SMN recruitment [11]. Following our previous papers, showing a mutual functional interaction between MYC and PRMT5 [27,28], in the present work we showed, for the first time, that MYC may have a role in transcript termination promoting PRMT5-dependent symmetrical dimethylation of RNAPII on R1810. The immunoprecipitations in Figure 1b indicate that PRMT5, SMN, MYC, and Omomyc—the latter widely used to interfere with MYC functions [21,23]—are associated with RNAPII. In this regard, MYC associates with several proteins that regulate RNAPII activity [16,61,62,63,64,65]. Further, RNAPII R1810 symmetrical dimethylation is not the only RNAPII post-translational modification affected by MYC. Indeed, we found that also Ser2 phosphorylation, which marks active transcripts elongation after the release of transcriptional pausing, is modulated by MYC and Omomyc. MYC-dependent induction of RNAPIISer2P was expected. However, our data indicate that Omomyc also may act on the transition between initiation and elongation by affecting Ser2P levels, depending on the relative amount of MYC, which is cell-context specific. In fact, in HEK293T cells, which possess a scarce amount of MYC molecules, Omomyc induced RNAPII-Ser2P (see Figure 3a, left panel). Conversely, in BT168FO cells, harboring elevated MYC levels, Omomyc caused RNAPII-Ser2P overall reduction (Figure 3b). Similar considerations may underlie the impact of Omomyc on MAX protein levels in cancer cells (Figure 3c). MYC overexpression in tumor cells causes a decrease in the formation of MAX/MAX homodimers, which, in normal conditions, attenuate the binding of MYC to specific (E-boxes) and non-specific DNA sequences [66,67,68]. Our findings that Omomyc is able to modulate the MYC/MAX ratio in cancer cells (Figure 3c) suggest that Omomyc may affect MYC/MAX-dependent transcription by binding to chromatin as homodimers that largely displace MYC from DNA and can compromise DNA binding of MAX/MAX complexes as well [23,24]. In this regard, Omomyc appears to function in a multiplicity of ways: modulating MYC and MAX DNA binding and protein levels, inducing changes of RNAPII methylation, and perturbing the MYC interactome [23,69] (see Figure 6 depicting a model of MYC and MYC inhibition impact on RNAPII activity).

When considering the whole transcriptome affected by MYC inhibition in glioblastoma stem cells, we found that gene clusters well established to be controlled by MYC activity were affected by Omomyc. The differential expression observed following Omomyc induction is expected to be largely the consequence of MYC inhibition. Nevertheless, our gene expression analyses—performed after relatively long perturbations of 24 and 4 h—cannot discriminate between direct and indirect effects. Furthermore, there is no consensus on the direct regulatory functions of MYC, as several authors argue that MYC is a transcriptional activator and repressor of selected gene subsets, whereas others have suggested that MYC acts as a general transcriptional amplifier [2,3,5,6,19]. On the other hand, our analysis of 100 direct MYC targets, the so-called *Muhar signature* [59], corroborates the hypothesis that Omomyc specifically represses the expression of authentic, direct MYC target genes.

At the chromatin level, the increased density at promoters may be explained by the consideration that Omomyc restrains transcriptional pause release and may thus cause an accumulation of RNAPII at promoters. The impaired R1810 symmetrical dimethylation, which affects termination and may lead to RNAPII accumulation at termination regions of active genes [11], may account for the increased density at terminators. However, we did not detect a striking link between RNAPII occupancy and changes in the expression of transcripts. Therefore, while Omomyc induction leads—either directly or indirectly—to global expression changes of MYC target genes in BT168FO cells, such changes cannot be solely explained by Omomyc-induced changes in RNAPII density at terminator and/or promoter sites, but would involve modulation by additional factors, the action of which might be directly influenced by Omomyc. In summary, Omomyc impairs MYC binding to promoters, perturbs the MYC interactome [23], and restrains MYC-dependent enhancement of transcriptional pause release and R1810 dimethylation. All this contributes to an adjustment of RNAPII distribution and to re-normalization of the expression of genes that are deregulated as a consequence of MYC overexpression—such as those governing the GSC phenotype [24]—by molecular mechanisms that are yet to be clarified.

## Figures and Tables

**Figure 1 biomedicines-11-00412-f001:**
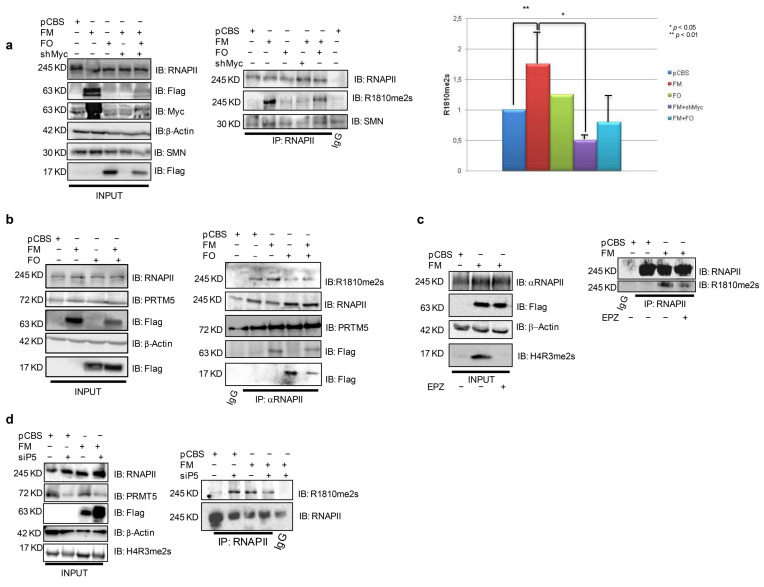
MYC ectopic expression enhances RNAPII R1810 symmetrical dimethylation via PRMT5. (**a**) Left: HEK293T cells transfected with pCBSFlagMYC, pCBSFlagOmomyc plasmids, and a shRNA against MYC, alone or in combination. Immunoprecipitations were performed by an RNAPII antibody. While ectopic MYC expression strongly increased RNAPII R1810 dimethylation, MYC shRNA and Omomyc inhibited such an increase (blots and densitometry histogram). (**b**): HEK293T cells transfected with pCBSFlagMYC and pCBSFlagOmomyc plasmids, alone or in combination. After 48 h, immunoprecipitations were performed by an RNAPII antibody. PRMT5, SMN, MYC, and Omomyc co-precipitated with RNAPII. (**c**) HEK293T: cells transfected with pCBSFlagMYC. The day after, cells were treated for 24 h with 5 μM EPZ01566 PRMT5 inhibitor or control vehicle; thereafter, immunoprecipitation was performed. EPZ015666 impaired symmetrical dimethylation of H4R3 and the MYC-dependent increase of R1810me2s and H4R3me2s, the latter was used as a control of EPZ activity. (**d**) HEK293T cells transfected with a smart pool siRNAs for PRMT5 (siP5) and after 24 h transfected with pCBSFlagMYC plasmid. After, additional 48 h immunoprecipitation was performed. siP5 blunts MYC-dependent increase of R1810me2s. H4R3me2s decrease was used as control of PRMT5-impaired activity in the presence of siP5. Each bar in the histogram represents mean ± SEM. Abbreviations. FM: FlagMYC; FO: FlagOmomyc; EPZ: EPZ015666; siP5: siPRMT5.

**Figure 2 biomedicines-11-00412-f002:**
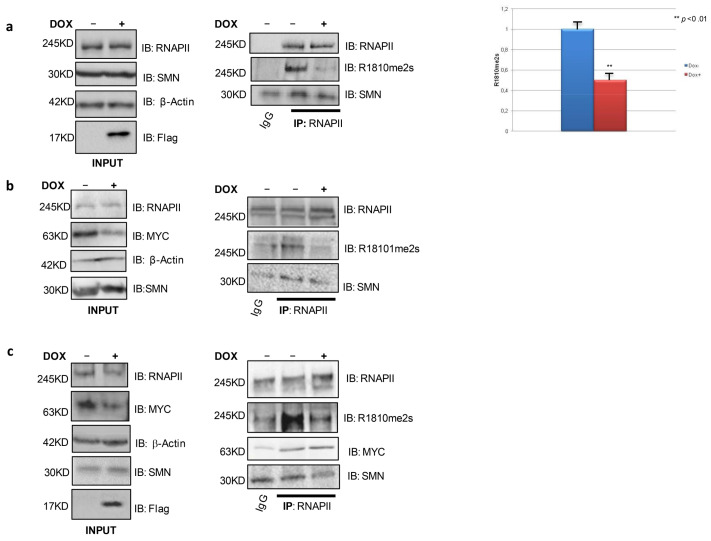
MYC inhibition by Omomyc and an shRNA against MYC affects RNAPII R1810 symmetrical dimethylation in cancer cells. Cells were treated with doxycycline for 24 h. Thereafter, immunoprecipitation was performed by means of an anti-RNAPII antibody, followed by immunoblotting with anti-RNAPII-R1810me2s, MYC, SMN, and RNAPII antibodies. (**a**) BT168FO cells. R1810 symmetrical dimethylation was severely impaired in the presence of FlagOmomyc. SMN binding to RNAPII also decreased. (**b**) BT168 cells infected with a lentivirus encoding a doxycycline-inducible shRNA against MYC. The shRNA strongly decreased RNAPII symmetrical dimethylation and SMN recruitment. (**c**) RamosFO cells. Also in this cancer cell line, MYC inhibition led to a decrease in RNAPII symmetrical dimethylation and SMN recruitment. Densitometry histograms are shown to the right of each panel. Each bar represents mean ± SEM.

**Figure 3 biomedicines-11-00412-f003:**
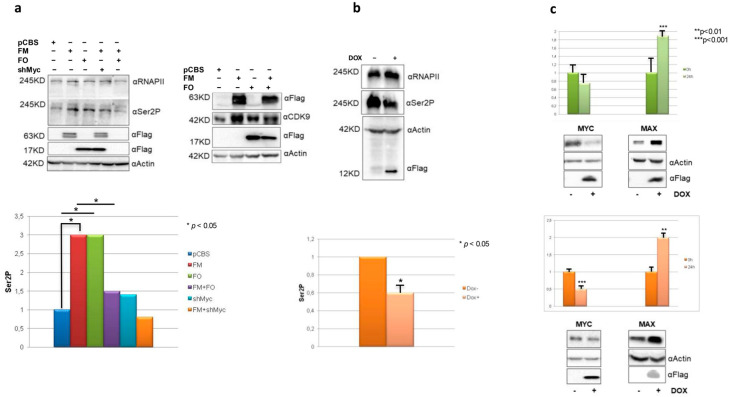
MYC inhibition impairs RNAPII Ser2 phosphorylation and increases Max protein. (**a**) Left: HEK293T cells transfected with pCBSFlagMYC, pCBSFlagOmomyc, and MYC shRNA plasmids—alone or in combination—and evaluated for RNAPIIser2 phosphorylation. Right: HEK293T cells transfected with pCBSFlagMYC, pCBSFlagOmomyc, and evaluated for CDK9 expression (**b**) BT168FO cells cultured in the presence or absence of doxycycline for 24 h, and probed with RNAPII Ser2P antibody. (**c**) BT168FO (top) and RamosFO cells (down) treated or not with doxycycline for 24 h, and probed with MAX and MYC antibodies.

**Figure 4 biomedicines-11-00412-f004:**
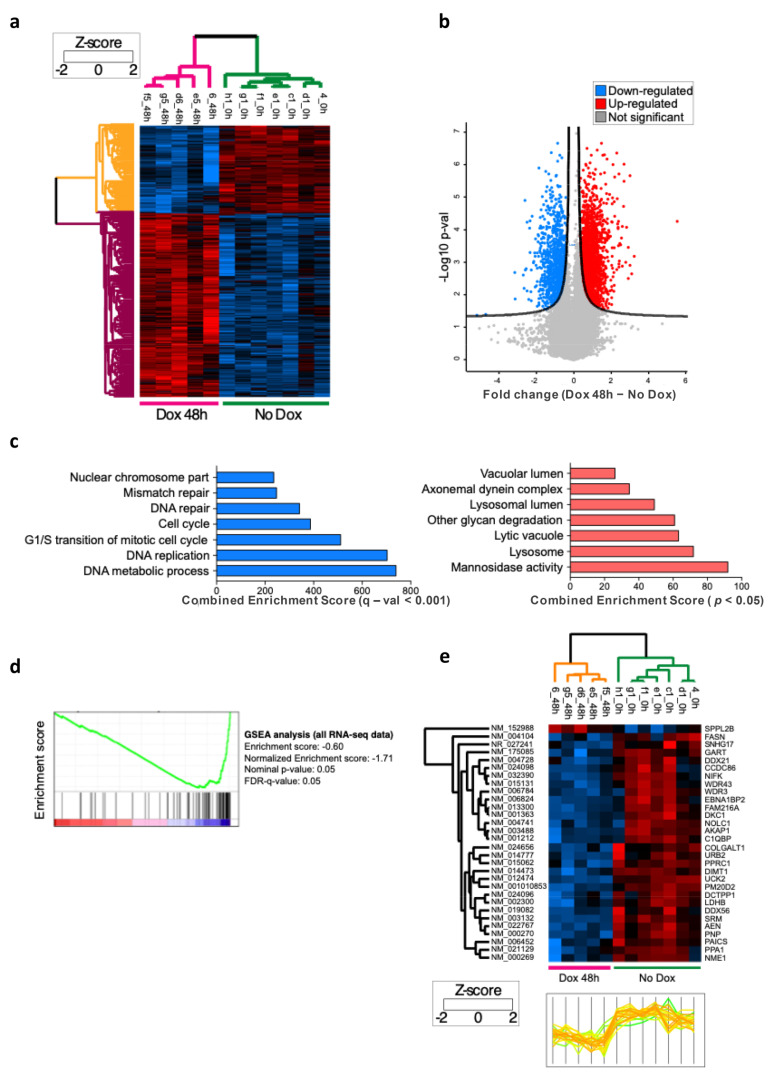
RNA sequencing and pathway analysis. (**a**,**b**) RNA sequencing analysis shows 2228 genes that are differentially expressed in BT168FO cells upon 48 h Omomyc induction: 1606 up-regulated and 622 down-regulated. (**c**) Functional enrichment analysis of GO terms in down-regulated (**left**) and up-regulated (**right**) genes upon 48 h Omomyc induction. The barplot shows the top enriched GO biological processes, cellular components, and molecular functions. (**d**) GSEA analysis with an enrichment score that confirmed the down-regulation of the Muhar signature (Muhar et al., 2018) [59] of direct MYC targets genes upon Omomyc induction. (**e**) Heatmap with profile plot of a subset of differentially expressed genes that are part of the Muhar signature. The subset includes genes strongly expressed—FPKM at least 10—and differentially expressed, at *p*-value threshold < 0.05. All of them, except one, were down-regulated upon Omomyc induction.

**Figure 5 biomedicines-11-00412-f005:**
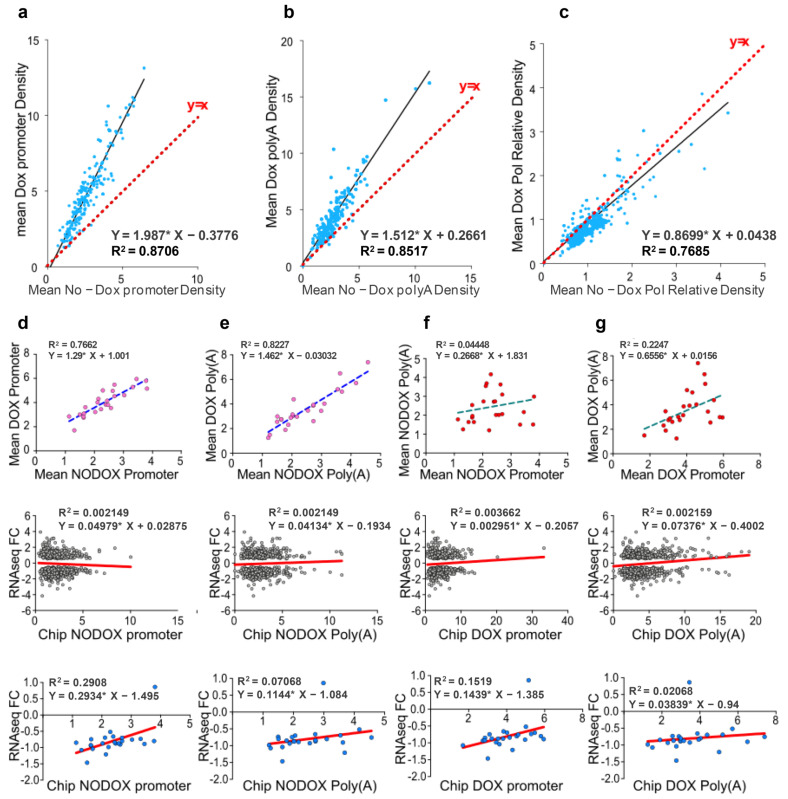
MYC inhibition affects RNAPII density at promoter and terminator regions. Correlation scatter plots of RNAPII occupancy and mRNA fold change (FC) in genes strongly expressed (FPKM ≥ 10) in at least one condition and with a significant Fold Change (*p*-value threshold < 0.05) in BT168FO cells treated or not with DOX for 48 h. **First two rows** (**a**–**g**): RNAPII density at promoter and PolyA regions; (**a**–**c**): all MYC target genes; (**d**–**g**): Muhar signature genes (the same as in Figure 4e). The scatter plots show an increased RNAPII occupancy upon Omomyc induction at both promoter and termination sites, slightly higher at promoters. **Third and fourth rows:** comparison between RNAseq Fold Change and CHIPseq density at promoters and terminators. **Third row:** all MYC targets; **fourth row:** Muhar signature genes (as in Figure 4).

**Figure 6 biomedicines-11-00412-f006:**
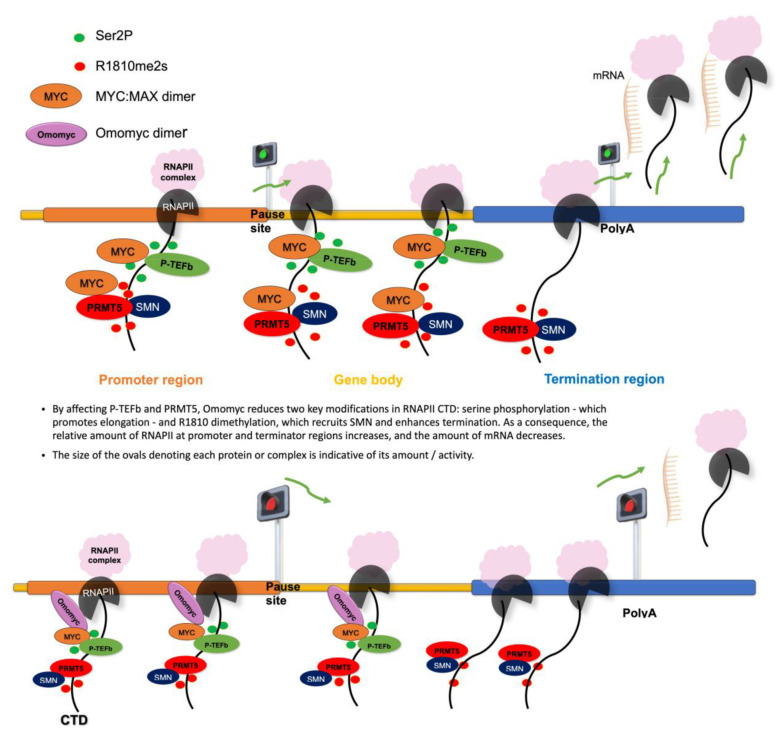
A model for MYC on RNAPII activity. Top: in the presence of high levels of MYC, RNAPII is widely distributed across regulators chromatin loci (both promoters and terminators) and gene bodies. Bottom: when MYC is inhibited, both pause-release and termination are rewired and RNAPII accumulates at promoters and transcription termination sites.

## Data Availability

ChIP-seq and RNA-seq data used in this study are accessible via Gene Expression Omnibus (GEO http://www.ncbi.nlm.nih.gov/geo/ accessed on 13 December 2022), with accession identifier GSE86519.

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
