# Peer review of "A New Insight into MYC Action: Control of RNA Polymerase II Methylation and Transcription Termination"

_biomedicines, 2023, doi:10.3390/biomedicines11020412_

Round 1

Reviewer 1 Report (Previous Reviewer 3)

With the authors' revision I have no pending questions.

Author Response

The authors wish to thank the Reviewer for her/his time spent in evaluating the updated version of our MS and for her/his positive evaluation.

Reviewer 2 Report (Previous Reviewer 2)

I would really appreciate the work done by the authors to improve the quality of the manuscript entitled “A new insight into MYC action: control of RNA polymerase II methylation and transcription termination”.

The authors carefully considered remarks given by all reviewers and provided a reliable explanation or even conducted additional experiments.

At this step I will address author’s responses to my previous comments:

Q1) Authors rewrite the abstract in way which make it very clear and reader-frendly. Now it is informative and provide all information which are important for readers.

Q2) Authors only partially address this drawback of their paper. Quality of blots is ok. Bar plots still are in low resolution (i.e. 1a, 2a-c right panel, 5). Please check if you are capable to increase it. Maybe it just a matter of conversion to pdf. You can reconsider it with Editor. Please treat this remark as care about quality of your presentation not negative feedback from my side.

Q3) Authors provide clear justification in their manuscript. I am only not sure about the general meaning and sense of phrase “biological and molecular layers”. Do authors think about both functional level and corresponding genetic background?

Q4) Corrected properly.

Q5) Authors refer to one of the image analysis-based methods to overcome some limitations of densidometry (PMID: 29522776). I can say that at this step it is acceptable.

Q6) Authors provided valuable discussion and corrected small errors related to Figure 6.

Minor: Authors should put information about morphology of cells in text of paper (just in the same way like they wrote in response for my revision). It is ok then.

In general, after allcorrectionsn, explanation and additional experiments I consider that this work meets standards to be published in Biomedicines.

Like underlined above some tiny changes might be introduced to manuscript to elevate in overall merit – nevertheless those changes are not obligatory.

I congratulate authors their work and whish many interesting results in the future.

Author Response

The authors wish to thank the Reviewer for the careful evaluation of our MS, both in the previous and revised form, and for her/his positive feedback.

As some minor concerns were raised, below are our point-to-point answers. Changes to the MS are tracked in red colour.

Q2) Authors only partially address this drawback of their paper. Quality of blots is ok. Bar plots still are in low resolution (i.e. 1a, 2a-c right panel, 5). Please check if you are capable to increase it. Maybe it just a matter of conversion to pdf. You can reconsider it with Editor. Please treat this remark as care about quality of your presentation not negative feedback from my side.

A2. We apologize for the scarce resolution of our bar plots. Now, we believe to have solved this issue as in figures 1a, 2a-c right panel and 3 (we believe the Reviewer meant bar plots of figure 3) of the newly revised MS.

Q3) Authors provide clear justification in their manuscript. I am only not sure about the general meaning and sense of phrase “biological and molecular layers”. Do authors think about both functional level and corresponding genetic background?

A3. We thank the Reviewer for this comment. In this phrase we wished to point pout that in the cells system we used in our experiments Myc acts similarly as a transcription factor ruling the activation of cancer-specific programs (molecular layer) which profoundly impact cancer cell behaviour (the biological layer). To better clarify this point, we changed the phrase at page 10, lines 321-323.

Minor: Q1. Authors should put information about morphology of cells in text of paper (just in the same way like they wrote in response for my revision). It is ok then.

A1. We thank the Reviewer for this comment. Information about cells morphology is now provided at page 10, lines 326-327 of the newly revised MS. 

Reviewer’s final comment: In general, after all corrections, explanation and additional experiments I consider that this work meets standards to be published in Biomedicines.

Like underlined above some tiny changes might be introduced to manuscript to elevate in overall merit – nevertheless those changes are not obligatory.

I congratulate authors their work and whish many interesting results in the future.

Authors’ response: We are extremely grateful to this Reviewer for her/his wonderful words which represent fuel to continue our Myc research projects in a country where funds for Basic Research are dramatically limited and difficult to obtain.

Reviewer 3 Report (New Reviewer)

The paper-A new insight into MYC action: control of RNA polymerase II 2 methylation and transcription termination- is a continuation of prior work on the interaction of PRMT5 with RNA polymerase II in the context of MYC action.  This article may have been transferred from another journal as the formatting suggests revisions based on prior comments.   The work shows that genes, particularly those associated with down-regulation based on the analysis by Muhar et al show accumulation of RNA poly II at promoters and at termination sites in response to Omomyc with expected down regulation.

I was a little confused until I checked back to find that Omomyc has a long track record in MYC research, and the group has been branching off from Omomyc work to look at PRMT5 and MYC interaction. Until that I wasn’t sure whether the paper was about MYC/RNA poly II or Omomyc.  Perhaps the introduction, despite being rewritten already, could use more modification to make the main point clearer.

The paper is limited in analytic power; however, it makes the point that PRMT5, SMN and MYC impact distribution of RNA poly II at both promoter and termination sites, in line with current literature about RNA pol II.  Why no mention of Omomyc impact in grafted tumors or the lung tumor paper?

Minor-In M&M-previously was spelled wrong.

Make sure the abbreviations section is complete-TSS

Not sure all antibodies are listed with catalog numbers.

I the results section, page 10, second paragraph-line in red type, “which in these cells----low MYC protein”; not sure what this means-please clarify.

Author Response

We wish to thank the Reviewer for having accepted to evaluate our MS in its revised form and for her/his positive evaluation. Below are our point-to point answers. Changes to the MS are tracked in red colour.

Major

Q1. I was a little confused until I checked back to find that Omomyc has a long track record in MYC research, and the group has been branching off from Omomyc work to look at PRMT5 and MYC interaction. Until that I wasn’t sure whether the paper was about MYC/RNA poly II or Omomyc.  Perhaps the introduction, despite being rewritten already, could use more modification to make the main point clearer.

A1. We thank the Reviewer for her/his time spent in taking information about our lab research history and thank the Reviewer for this comment. We have modified again the Introduction section, as at page 2, line 73-84 of the newly revised version of the MS.

Q2. The paper is limited in analytic power; however, it makes the point that PRMT5, SMN and MYC impact distribution of RNA poly II at both promoter and termination sites, in line with current literature about RNA pol II.  Why no mention of Omomyc impact in grafted tumors or the lung tumor paper?

A2. We thank the Reviewer for this comment. As for the previous request, mention of Omomyc impact on a variety of cancer types has been provided in the introduction section, page 2, lines 73-84. However, since this study was funded by the an Italian Association for Cancer Research grant on glioblastoma multiforme, the experiments on the role of Myc in transcription termination were limited to glioblastoma stem cells and RAMOS cells as another paradigm of Myc-overexpressing cancer cells.

Minor

Q1. In M&M-previously was spelled wrong

A1. We thank the Reviewer for this observation. We have checked and corrected.

Q2. Make sure the abbreviations section is complete-TSS

A2. We thank the Reviewer. Now, the abbreviations section is complete.

Q3. Not sure all antibodies are listed with catalog numbers.

A3. We have checked and added Abs catalogue numbers where missing

Q4. In the results section, page 10, second paragraph-line in red type, “which in these cells----low MYC protein”; not sure what this means-please clarify.

A4. We thank the Reviewer for having pointed our attention to this issue. In this paragraph we underlie that RNAPII symmetrical dimethylation is constitutively present in BT168 glioblastoma stem cells and Ramos cells, due to the high levels of Myc. Therefore, the only way we had to evaluate Myc impact on RNAPIIme2s was to blunt its activity by Omomyc. Conversely, in HEK293T cells, which are characterized by low Myc levels, we were able to show the direct Myc activity (depending on PRMT5) on RNAPIIme2s, by overexpression of a Flag-tagged Myc plasmid (previous paragraph in the result section, pages 7-10). We have rewritten the phrase indicated, better clarifying the issue raised by the Reviewer.

This manuscript is a resubmission of an earlier submission. The following is a list of the peer review reports and author responses from that submission.

Round 1

Reviewer 1 Report

In the study titled " A new insight into MYC action: control of RNA polymerase II methylation and transcription termination" , the authors reported that MYC produces an increase in R1810me2s. Since it is known that MYC regulates PRMT5 and PRMT5 catalyzes R1810me2s, the novelty of their finding is limited. And the data they present needs to be improved.

Some concerns

1.Since they already applied shRNA for PRMI5 in figure 1a, a result to show that if shRNA for PRMI5 could have the same effect as EPZ015666 in figure 1c would be better to verify that PRMT5 was required for MYC induced R1810me2s.

2. Western blot is the most important technique throughout the article. However, the Western blot is of poor quality.. I can not accepted those nasty bands… For example, the bolts for R1810me2s in Figure1a middle blots look like an uncompleted membrane transferring rather than reducing expression. Figure 1a and 1b are the same data from repeated experiments. The authors need to well organize their results.

3. Omomyc is a well-known MYC inhibitor. I do not understand the novelty for the authors to use RNAseq only to prove Omomyc specifically influences the expression of genes and pathways that are regulated by MYC.

4. The introduction is not properly organized. It might be separated into three or four paragraphs, one for Myc, one for RNAPII and PRMI5, and so on to summarize the author's findings. That would be preferable to the readers.

5. The manuscript lacks a discussion section.

Reviewer 2 Report

Authors of the work „A new insight into MYC action: control of RNA polymerase II methylation and transcription termination” address its impact on RNAPII functioning.  In general, study is well designed and relatively complex which I consider being a positive aspect of this work.

In general, the abstract is well written – nevertheless, it is a little bit chaotic. Authors might think about making it more clear to the reader – it is not obligatory. It is good to give it to read to someone not familiar with your study and ask if everything is clear.

Major issues:

1.       Quality of figures must be increased. The resolution of images is very low – it creates a negative impression for the reader. Please increase resolution (i.e. 300 or 600 dpi.) and increase the thickness of lines in bar plots etc.

2.       Could authors justify more deeply why they choose BT168 and RAMOS cells?

3.       Please correct the proper placement of the caption Figuresgure 2,3,5.  Writing about statistical significance is a little bit chaotic. Once written in a figure description I find it not necessary to write it on the figure.

4.       Please justify the use of densitometry of western blots to compare protein abundance. It should be underlined that this method is to some extent considered controversial for quantitative comparison of western blotting results. (i.e. PMID 19517440, 30800670 etc..)

5.       In fact authors discuss their work together with showing results that are acceptable. Nevertheless, they must add a summary/conclusion section to make the manuscript clear. This part has to be improved – importantly authors should describe figure 6 (there is still lacking of any link to this figure in text)

Minor:

It will be also beneficial to add supplementary figure showing cells morphology for particular conditions described in study. 

Reviewer 3 Report

Although I have no expertise to evaluate the methodological details of this work in depth, I consider that the authors contributed with an investigation of fundamental importance to extend the meaning of the MYC action in tumor cells.

I suggest English revision of the main text and of the non-published material to improve the quality of the article and  that the authors insert a topic in which the large number of abbreviations cited in the text also appear referred in full.